# Metabolomics and network pharmacology-based identification of phenolic acids in *Polygonatum kingianum* var. *grandifolium* rhizomes as anti-cancer/Tumor active ingredients

**Xiaolin Wan, Lingjun Cui, Qiang Xiao** *

Hubei Key Laboratory of Biological Resources Protection and Utilization (Hubei Minzu University), Enshi, China

* 1992022@hbmzu.edu.cn

**Data Availability Statement:** All relevant data are included in the manuscript file.

## Abstract

Broadly targeted metabolomics techniques were used to identify phenolic acid compounds in *Polygonatum kingianum* var. *grandifolium* (PKVG) rhizomes and retrieve anti-cancer/tumor active substance bases from them. We identified potential drug targets by constructing Venn diagrams of compound and disease targets. Further, KEGG pathway analysis was performed to reveal the relevant pathways for anti-cancer/tumor activity of PKVG. Finally, we performed molecular docking to determine whether the identified proteins were targets of phenolic acid compounds from PKVG rhizome parts. The study's results revealed 71 phenolic acid compounds in PKVG rhizomes. Among them, three active ingredients and 42 corresponding targets were closely related to the anticancer/tumor activities of PKVG rhizome site phenolic acids. We identified two essential compounds and eight important targets by constructing the compound-target pathway network. 2 essential compounds were androsin and chlorogenic acid; 8 key targets were MAPK1, EGFR, PRKCA, MAPK10, GSK3B, CASP3, CASP8, and MMP9. The analysis of the KEGG pathway identified 42 anti-cancer/tumor-related pathways. In order of degree, we performed molecular docking on two essential compounds and the top 4 targets, MAPK1, EGFR, PRKCA, and MAPK10, to further validate the network pharmacology screening results. The molecular docking results were consistent with the network pharmacology results. Therefore, we suggest that the phenolic acids in PKVG rhizomes may exert anti-cancer/tumor activity through a multi-component, multi-target, and multi-channel mechanism of action.

## Introduction

The "Ming Yi Bie Lu" first published the *Polygonatum*, a perennial herb of the *Liliaceae* with the same origins as food and medicine. The 2020 edition of the Chinese Pharmacopoeia lists the Chinese medicine *Polygonatum*'s basal plants as *Polygonatum sibiritum* Red, *Polygonatum*

**Funding:** This study was jointly funded by the National Natural Science Foundation of China [NSFC], grant number 31260057, QX], The Natural Science Foundation of Hubei Province (Joint Fund) [NSFHP, grant number 2023AFD077, QX], Open Fund of Hubei Key Laboratory of Biological Resources Protection and Utilization (Hubei Minzu University) [OHBP, grant number KYPT012403, QX], and Major Special Project of Technological Innovation of Hubei Provincial Science and Technology Department [MHT], grant number 2019ACA120, QX]. The funders had no role in study design, data collection and analysis, decision to publish, or preparation of the manuscript.

**Competing interests:** The authors have declared that no competing interests exist.

*kingianum* coll.et Hemsl and *Polygonatum cyrtonema* Hua. The world distributes *Polygonatum*, including in China, Japan, Korea, India, Russia, Europe, and North America [1, 2]. *Polygonatum kingianum* var. *grandifolium* (PKVG), a variant of *Polygonatum kingianum* coll.et Hemsl, have garnered attention due to their large production volume and frequent use in folk medicine as a source of *Polygonatum* herbal medicine [3]. Previous studies have found that *Polygonatum* contain polysaccharides, saponins, flavonoids, lignans, amino acids, phenolic acids, alkaloids, and a variety of trace elements [4]. These compounds have demonstrated significant anti-fatigue, anti-aging, metabolic modulation, immunomodulation, anti-inflammatory, neuroprotective, anti-diabetic, and anti-cancer effects [5]. These compounds and pharmacological effects emphasize *Polygonatum*'s potential.

Traditional Chinese medicines' therapeutic effects often stem from the synergistic effects of their multiple components, so it is critical to explore the role of their chemical components in depth. The key to achieving this research goal is constructing a scientific and efficient analytical tool. Broadly targeted metabolomics analysis is a new metabolomics approach. It combines the advantages of targeted metabolomics (high sensitivity) and non-targeted metabolomics (high throughput). It is distinguished by high throughput, ultra-sensitivity, and comprehensive coverage of metabolites, allowing it to analyze thousands of metabolites in plant samples in a single step, qualitatively and quantitatively [6]. Many fields use broadly targeted metabolomics technology, including agriculture, medicine, and food science. This study used broadly targeted metabolomics technology for rapid qualitative and quantitative analysis of chemical constituents in PKVG rhizomes. The study's results can provide a reference for the quality control of PKVG rhizomes and a scientific basis for the study of the medicinal efficacy of PKVG rhizomes.

Phenolic acids are one of the most prevalent classes of bioactive chemicals of plant origin. They are found in large quantities in foods such as berries [7], nuts [8], and whole grains [9]. Phenolic acids have been reported to reduce the adhesion of various microbial cells, ultimately reducing the etiology of disease [10]. Other studies have shown that phenolic acids have received widespread attention for their potential antioxidant, cardioprotective, anti-inflammatory, anti-atherosclerotic, immunomodulatory, anti-allergic, anti-thrombolytic, antimicrobial, antitumor, anti-obesity, anticancer, and antidiabetic properties [11–13]. Currently, cancer has been recognized as the second-leading cause of death worldwide. Conventional methods for the treatment of cancer include chemotherapy or radiotherapy. However, these methods are usually associated with various side effects and many drawbacks in clinical practice [14]. With the rapid development of bioinformatics technology, cyber pharmacology has become a powerful tool for exploring the mechanisms of action of herbal medicines. It provides a systematic understanding of drug action and disease complexity and models constructed based on cyber pharmacology help to elucidate the role of herbal medicines in specific diseases [15].

Therefore, the present study combined metabolomics and network pharmacology to elucidate the anticancer/antitumor mechanism of phenolic acids in PKVG rhizomes. It provides an important basis for the study of new anticancer/antitumor drugs.

## Materials and methods

### Plant materials

The test sample was a three-year-old PKVG rhizome. We planted the PKVG at the test site of the College of Forestry and Horticulture, Hubei Minzu University. The bottom soil consisted of vermiculite and peat soil (8:1 ratio). We collected three biological replicates of PKVG rhizomes. We chopped, mixed, and loaded the sampled PKVG rhizomes into sterile centrifuge tubes, snap-frozen them in liquid nitrogen, and stored them at -80°C until we needed them.

## Sample preparation and extraction

Samples were prepared and extracted using methods provided by Metware Biotechnology Ltd. (Wuhan, China). The data acquisition instrumentation system mainly consisted of ultra-performance liquid chromatography (UPLC) (SHIMADZU Nexera X2) and tandem mass spectrometry (MS/MS) (Applied Biosystems 4500 QTRAP).

Liquid phase conditions: (i) column: Agilent SB-C18 1.8 μm, 2.1 mm * 100 mm; (ii) mobile phases: ultra-pure water (with 0.1% formic acid added) for phase A and acetonitrile (with 0.1% formic acid added) for phase B; (iii) the proportion of B-phase was 5% for wash 0.00 min, and then the proportion of B-phase was increased linearly to 95% within 9.00 min and was maintained at 95% for 1 min; from 10.00 to 11.10 min, the B-phase ratio decreased to 5% and equilibrated at 5% for 14 min; (iv) flow rate: 0.35 mL/min; column temperature: 40˚C; injection volume: 4 μL. UPLC-MS/MS was carried out by Metware Biotechnology Ltd. (Wuhan, China). Metabolomics data were obtained in electrospray ionization negative (ESI-) and positive (ESIIC) modes. The ion spray voltage was -4500 V for ESI- and 5500 V for ESIC; the ion source gas I (GSI), gas II (GSII), and curtain gas (CUR) were set to 50, 60, and 25 psi, respectively; the collision-induced ionization parameter was set to high; and the electrospray ionization source (ESI) temperature was 550˚C.

## Qualitative and quantitative metabolite analysis

Raw data were initially converted to mzXML format using Proteo Wizard. Subsequently, peak extraction, alignment, and retention time correction were performed using XCMS software. To ensure data quality, peaks with a missing rate exceeding 50% in any sample group were excluded, and missing values were imputed through the KNN method. The peak areas were further adjusted using the SVR method. Metabolites were identified by matching the processed peaks to the proprietary database of Metware Biotechnology Company (Wuhan, China), as well as public and predictive libraries, following the MetDNA approach. Only metabolites with a composite identification score $\geq 0.5$ and a coefficient of variation (CV) $<0.3$ in QC samples were retained for subsequent analysis. Positive and negative ion modes were integrated to retain compounds with the highest identification confidence and the lowest CV values, generating a comprehensive data file for all samples. Metabolites were characterized using both primary and secondary MS data, and their relative abundance across samples was determined based on the chromatographic peak areas.

In order to compare the differences in the content of each metabolite among all the detected metabolites in different samples, based on the information of metabolite retention time and peak shape, we corrected the mass spectrometry peaks of each metabolite detected in different samples to ensure the accuracy of qualitative and quantitative.

## Data collection on active components and targets of phenolic acid compounds from PKVG rhizomes

We used the TCMSP database (https://old.tcmsp-e.com/tcmsp.php) to search for PKVG rhizome phenolic acid analog drug constituents. OB $\geq 10\%$ and DL $\geq 0.1$ [16] were used as the screening conditions to obtain the active ingredients of PKVG rhizome phenolic acid analogs and their corresponding targets, and the UniProt database (https://www.uniprot.org/) was used for canonical target naming. Finally, we used Cytoscape 3.9.1 software to make a network diagram of the active ingredients and targets of PKVG rhizome phenolic acid analogs.

## Cancer/tumor target information collection

In the GeneCards database (https://www.genecards. org/), we searched for relevant targets by entering "cancer/tumor". The Wayne plots of the active ingredient targets of PKVG rhizome

phenolic acid analogs and cancer/tumor targets were plotted using the online software Venny 2.1.0 (http://www.liuxiaoyuyuan.cn/) to obtain PKVG rhizome phenolic acid analogs-cancer/tumor common targets, which were predicted to be the potential PKVG rhizome phenolic acid analogs treatment for cancer/tumor. The target was predicted to be a potential target for the treatment of cancer/tumor.

### Gene Ontology (GO) and Kyoto Encyclopedia of Genes and Genomes (KEGG) analyses

The common targets of PKVG rhizome phenolic acids and cancer/tumor were entered into the DAVID database (https://david.ncifcrf.gov/tools.jsp/) platform, and the species was limited to humans. GO enrichment analysis and KEGG pathway annotation analysis of potential targets of PKVG rhizome phenolic acids for cancer/tumor treatment were performed to screen important signaling pathways of phenolic acids in PKVG rhizomes for cancer/tumor treatment. The GO function enrichment histogram and KEGG signaling pathway enrichment bubble diagram were drawn in R software from the analysis results.

### Network construction and analysis

The compound-target-pathway network was constructed using Cytoscape 3.9.0. In the network, compounds, targets, and pathways were represented by nodes, and interactions between two nodes were represented by edges. In addition, the importance of each node in the networks was evaluated using a crucial topological parameter, namely degree. The topological properties were analyzed using the Network Analyzer plug-in for Cytoscape 3.9.0 to confirm the key components and targets, such as degree centrality (DC), betweenness centrality (BC), and closeness centrality (CC).

### Molecular docking techniques and visualization

The 3D structures of the active components of PKVG rhizome phenolic acid analogs were downloaded from the PubChem database (https://pubchem.ncbi.nlm.nih.gov/), and the receptor proteins corresponding to the key targets were downloaded from the RCSB PDB database (https://www.rcsb.org/). The sdf files were converted into the pdb file format using Discovery Studio 2019 software. PyMOL 4.3.0 software was used to separate the original ligand and protein structures and to dehydrate and remove the organics. Furthermore, AutoDockTools-1.5.7 software was used to process the proteins as follows: non-polar hydrogen was added, the Gasteiger charge was calculated, the AD4 type was assigned, and the flexible bonds of small molecules/ligands were set to be rotatable. Based on the original ligand coordinates, the docking box was adjusted to include all protein structures. Furthermore, the receptor protein was set to a semiflexible butt joint, and the Lamarckian genetic algorithm was selected. The docking results were obtained by running autogrid4 and autodock4; as a result, the binding energies were revealed. Finally, visualization and analysis were performed by PyMOL 4.3.0 software.

## Results

### Detection of phenolic acid metabolites and differential metabolite analysis of PKVG rhizomes

In order to identify and better understand the phenolic acid metabolites in PKVG rhizomes in more detail, we performed a broadly targeted metabolomic analysis of PKVG rhizomes. A total of 71 phenolic acids were identified (Table 1). As can be seen from Table 1, a total of 19 phenolic acids were detected in the positive ion mode and 52 in the negative ion mode. In addition, the top 5 phenolic acid compounds in PKVG rhizomes were 4-nitrophenol,

**Table 1. UPLC-MS/MS results of phenolic acid compounds in PKVG rhizome extracts.**

| Peak number | Index | Molecular weight (Da) | Formula | Ionization model | Compounds | Measured excimer ion peak (m/z) |
|---|---|---|---|---|---|---|
| 1 | pme2828 | 139.0269 | $C_6H_5NO_3$ | +H | 4-Nitrophenol | 34271160.42 |
| 2 | Lmxp011770 | 278.1518 | $C_{16}H_{22}O_4$ | +H | Diisobutyl phthalate | 13791107.00 |
| 3 | Lmlp012720 | 278.1518 | $C_{16}H_{22}O_4$ | +H | Dibutyl phthalate | 13090094.91 |
| 4 | Lmwp011196 | 390.277 | $C_{24}H_{38}O_4$ | +H | Bis (2-ethylhexyl) phthalate | 3976017.85 |
| 5 | Lmmp010562 | 390.277 | $C_{24}H_{38}O_4$ | +H | Diisooctyl Phthalate | 3900247.07 |
| 6 | pmn001367 | 316.0794 | $C_{13}H_{16}O_9$ | -H | Protocatechuic acid-4-O-glucoside | 2894640.82 |
| 7 | pmb2871 | 316.0794 | $C_{13}H_{16}O_9$ | -H | 1-O-Gentisoyl-β-D-glucoside | 1797950.01 |
| 8 | Lmbp000728 | 120.0575 | $C_8H_8O$ | +H | (S)-2-Phenyloxirane | 1089991.67 |
| 9 | NK10264324 | 126.0317 | $C_6H_6O_3$ | +H | Phloroglucinol; 1,3,5-Benzenetriol | 1072580.24 |
| 10 | pmb3142 | 300.0845 | $C_{13}H_{16}O_8$ | -H | Salicylic acid-2-O-glucoside | 1029527.16 |
| 11 | Zmhn005413 | 336.0481 | $C_{15}H_{12}O_9$ | -H | p-Dimeric galloyl methyl ester | 1023436.25 |
| 12 | pmb3107 | 360.1057 | $C_{15}H_{20}O_{10}$ | -H | Glucosyringic Acid | 900840.02 |
| 13 | NK10253223 | 167.0582 | $C_8H_9NO_3$ | +H | 2-Amino-3-methoxybenzoic acid | 650870.58 |
| 14 | Lmyn000160 | 238.0689 | $C_8H_{14}O_8$ | -H | Mucic acid Dimethyl Ester | 487746.78 |
| 15 | pmp001285 | 148.016 | $C_8H_4O_3$ | +H | Phthalic anhydride | 483762.15 |
| 16 | Jmbp006554 | 388.1158 | $C_{20}H_{20}O_8$ | +H | Ethyl Rosmarinate | 432962.91 |
| 17 | pmn001681 | 166.0994 | $C_{10}H_{14}O_2$ | -H | 1-(4-Methoxyphenyl)-1-propanol | 387652.01 |
| 18 | pmb2654 | 461.1533 | $C_{19}H_{27}NO_{12}$ | -H | Anthranilate-1-O-Sophoroside | 375673.28 |
| 19 | pmb2497 | 198.0528 | $C_9H_{10}O_5$ | -H | 4-Hydroxy-3-methoxymandelate | 357663.15 |
| 20 | Zmhn001926 | 300.084 | $C_{13}H_{16}O_8$ | -H | 1-O-Salicyloyl-β-D-glucose | 206817.47 |
| 21 | pmn001518 | 332.0743 | $C_{13}H_{16}O_{10}$ | -H | 1-O-Galloyl-β-D-glucose | 203521.01 |
| 22 | Zmhn002422 | 356.1102 | $C_{16}H_{20}O_9$ | -H | 1-O-Feruloyl-β-D-glucose | 201251.17 |
| 23 | Hmqn000843 | 302.0998 | $C_{13}H_{18}O_8$ | -H | Tachioside | 192752.66 |
| 24 | pmn001519 | 336.0481 | $C_{15}H_{12}O_9$ | -H | Galloyl Methyl gallate | 166514.75 |
| 25 | mws0458 | 152.0474 | $C_8H_8O_3$ | -H | Vanillin; 4-Hydroxy-3-Methoxybenzaldehyde | 166341.69 |
| 26 | Zmhn002301 | 326.0996 | $C_{15}H_{18}O_8$ | -H | p-Coumaric acid-4-O-glucoside | 154949.25 |
| 27 | pmb0069 | 121.0528 | $C_7H_7NO$ | +H | Benzamide | 152656.67 |
| 28 | pme2598 | 168.042 | $C_8H_8O_4$ | -H | 3,4-Dihydroxybenzeneacetic acid | 147197.50 |
| 29 | pmn001419 | 326.1002 | $C_{15}H_{18}O_8$ | -H | 1-O-p-Coumaroyl-β-D-glucose | 145298.75 |
| 30 | Cmsp002787 | 226.0841 | $C_{11}H_{14}O_5$ | +H | 3,4'-Dihydroxy-3',5'-dimethoxypropiophenone | 135043.55 |
| 31 | Hmhn003067 | 326.1002 | $C_{15}H_{18}O_8$ | -H | Phenylpropionic acid-O-β-D-glucopyranoside | 110752.27 |
| 32 | MWSslk097 | 194.0579 | $C_{10}H_{10}O4$ | +H | Dimethyl phthalate | 105455.28 |
| 33 | Lmbn013410 | 206.1671 | $C_{14}H_{22}O$ | -H | 2,4-Di-Tert-Butylphenol | 101110.72 |
| 34 | Lmln010063 | 206.1671 | $C_{14}H_{22}O$ | -H | 2,6-Di-tert-butylphenol | 98167.34 |
| 35 | Lmtn002233 | 328.1158 | $C_{15}H_{20}O_8$ | -H | Androsin | 80932.86 |
| 36 | MWSmce248 | 164.0473 | $C_9H_8O_3$ | -H | 3-Hydroxcinnamic Acid | 80856.59 |
| 37 | Zmhn002227 | 386.1208 | $C_{17}H_{22}O_{10}$ | -H | 4-O-Glucosyl-sinapate | 66306.75 |
| 38 | Wmmp000182 | 302.0063 | $C_{14}H_6O_8$ | -H | ellag icacid | 65552.68 |
| 39 | pme3437 | 212.0685 | $C_{10}H_{12}O_5$ | +H | Eudesmic acid (3,4,5-trimethoxybenzoic acid) | 62081.79 |
| 40 | Lmtn002324 | 432.1632 | $C_{19}H_{28}O_{11}$ | -H | Benzyl-(2"-O-glucosyl) glucoside | 60778.40 |
| 41 | pme0422 | 194.0579 | $C_{10}H_{10}O_4$ | -H | Isoferulic Acid | 59409.52 |
| 42 | HJN003 | 386.1219 | $C_{17}H_{22}O_{10}$ | -H | 1-O-Sinapoyl-β-D-glucose | 55351.41 |
| 43 | mws0014 | 194.0579 | $C_{10}H_{10}O_4$ | -H | Ferulic acid | 55349.13 |
| 44 | Lmyn003028 | 432.1632 | $C_{19}H_{28}O_{11}$ | -H | Benzyl-β-gentiobioside | 53797.74 |
| 45 | MWSmce328 | 516.1268 | $C_{25}H_{24}O_{12}$ | +H | Isochlorogenic acid C | 51053.43 |
| 46 | Yshj000011 | 332.1107 | $C_{14}H_{20}O_9$ | +H | 2,6-dimethoxy-4-hydroxyphenyl-1-O-beta-D-glucopyranoside | 47820.04 |

*(Continued)*

**Table 1.** (Continued)

| Peak number | Index | Molecular weight (Da) | Formula | Ionization model | Compounds | Measured excimer ion peak (m/z) |
|---|---|---|---|---|---|---|
| 47 | Zmhn001793 | 342.0956 | $C_{15}H_{18}O_9$ | -H | 6-O-Caffeoyl-D-glucose | 44859.41 |
| 48 | mws0628 | 122.0368 | $C_7H_6O_2$ | -H | 4-Hydroxybenzaldehyde | 43175.01 |
| 49 | pmn001382 | 516.1268 | $C_{25}H_{24}O_{12}$ | -H | Isochlorogenic acid A | 43124.62 |
| 50 | Li512115 | 516.1268 | $C_{25}H_{24}O_{12}$ | -H | Isochlorogenic acid B | 42291.21 |
| 51 | Lman002731 | 342.0951 | $C_{15}H_{18}O_9$ | -H | Grevilloside F | 40605.37 |
| 52 | pmn001315 | 458.1424 | $C_{20}H_{26}O_{12}$ | -H | Regaloside | 36128.30 |
| 53 | pmb3061 | 500.153 | $C_{22}H_{28}O_{13}$ | -H | 5-O-p-Coumaroylquinic acid O-glucoside | 33770.19 |
| 54 | Lmmn001643 | 164.0473 | $C_9H_8O_3$ | -H | 2-Hydroxycinnamic acid | 30989.15 |
| 55 | Hmtn001302 | 300.0851 | $C_{13}H_{16}O_8$ | -H | Glucosyloxybenzoic acid | 28627.07 |
| 56 | Lmzn001983 | 376.1369 | $C_{16}H_{24}O_{10}$ | -H | D-Threo-guaiacylglycerol-7-O-β-D-glucoside | 26420.43 |
| 57 | pmb3064 | 500.153 | $C_{22}H_{28}O_{13}$ | -H | 3-O-p-Coumaroylquinic acid-O-glucoside | 25851.64 |
| 58 | pmn001690 | 328.1522 | $C_{16}H_{24}O_7$ | -H | 3-Hydroxy-4-isopropylbenzylalcohol-3-O-glucoside | 24789.39 |
| 59 | MWSslk066 | 168.0423 | $C_8H_8O_4$ | +H | 3-Hydroxy-4-methoxybenzoic acid; Isovanillic Acid | 23066.94 |
| 60 | Lmmn000774 | 344.1107 | $C_{15}H_{20}O_9$ | -H | Dihydro caffeoyl glucose | 18421.04 |
| 61 | Lmjp003731 | 530.1424 | $C_{26}H_{26}O_{12}$ | +H | 3,4-O-Dicaffeoylquinic Acid Methyl Ester | 15812.98 |
| 62 | Lmjp003822 | 530.1424 | $C_{26}H_{26}O_{12}$ | +H | 3,5-O-Dicaffeoylquinic Acid Methyl Ester | 15812.98 |
| 63 | pme3443 | 208.0736 | $C_{11}H_{12}O_4$ | -H | Sinapinaldehyde | 15394.70 |
| 64 | pmn001525 | 478.1475 | $C_{23}H_{26}O_{11}$ | -H | 3,5-Digalloylshikimic acid | 9948.08 |
| 65 | Jmwn002117 | 332.1107 | $C_{14}H_{20}O_9$ | -H | 2-(3,4-dihydroxyphenyl) ethanediol 1-O-β-D-glucopyranoside | 9256.81 |
| 66 | Lmmn001294 | 332.1107 | $C_{14}H_{20}O_9$ | -H | Koaburaside | 9256.81 |
| 67 | pmn001511 | 286.1053 | $C_{13}H_{18}O_7$ | -H | 3-Hydroxy-5-Methylphenol-1-O-Glucoside | 8358.38 |
| 68 | mws0178 | 354.0951 | $C_{16}H_{18}O_9$ | -H | Chlorogenic acid (3-O-Caffeoylquinic acid) | 7336.56 |
| 69 | Lmbn004847 | 180.0786 | $C_{10}H_{12}O_3$ | -H | 4-Methoxyphenylpropionic acid | 6560.65 |
| 70 | MWS1839 | 166.063 | $C_9H_{10}O_3$ | -H | Ethylparaben | 4314.38 |
| 71 | MWS2070 | 180.0786 | $C_{10}H_{12}O_3$ | -H | Propyl 4-hydroxybenzoate | 3289.45 |

diisobutyl phthalate, dibutyl phthalate, bis (2-ethylhexyl) phthalate, and diisooctyl phthalate, and all 5 compounds were detected in the positive ion mode.

## Target recognition and disease mapping

Targeted identification of 71 constituents was performed using the TCMSP database. A total of 22 chemical components of TCM were identified. To further identify the key active ingredients, we used OB ≥ 10% and DL ≥ 0.1 as screening criteria. As a result, three compounds were identified among the 22 metabolites. These three phenolic acids were diisooctyl phthalate, androsin, and chlorogenic acid, and the analysis of their disease targets showed that 97 targets were obtained after removing duplicates. Then, 2626 targets were obtained from the Genecards database by entering "cancer/tumor" as a keyword. The analysis was performed using Venny 2.1.0 online software to identify common compound targets and disease targets (Fig 1). Forty-two targets were identified in terms of anti-cancer/tumor activity.

## GO and KEGG pathway enrichment analysis

The 42 common targets of phenolic acids and cancer/tumor were subjected to GO functional enrichment analysis using the DAVID platform. The GO functional enrichment analysis

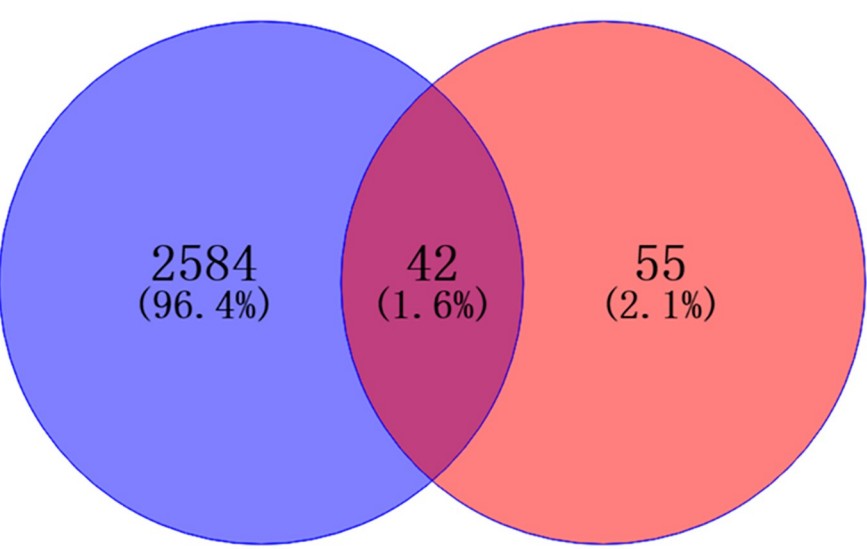

**Fig 1. Venn diagram of antitumor activity of PKVG rhizome phenolic acid analogs.**

yielded 128 BP, which were filtered according to P-value and found to be mainly associated with proteolysis, collagen catabolic process, and extracellular matrix disassembly. The analysis yielded 21 CC entries, which were filtered according to P-value and were mainly associated with the cytosol, cytoplasm, and extracellular matrix. The analysis yielded 39 entries for MF, which were filtered based on P-value and were mainly associated with endopeptidase activity, peptidase activity, and metalloendopeptidase activity (Fig 2).

Forty-two targets were enriched with 86 pathways associated with anti-cancer/tumor effects, of which 42 pathways were closely related to cancer/tumor. Fig 3 shows the results of the KEGG pathway analysis of anti-cancer/tumor effects of PKVG rhizomes. The results showed that pathways in cancer, pathways of neurodegeneration-multiple diseases, and the MAPK signaling pathway were the major pathways involved in the anti-cancer/tumor effects of PKVG rhizomes. Pathways in cancer had the largest number of Pathways in cancer have the largest number of anti-cancer/tumor-associated targets, including EGLN1, GSK3B, MMP1, MMP2, PRKCA, MMP9, EGFR, MAPK10, AR, CASP7, CASP8, CASP3, and MAPK1. Pathways of neurodegeneration-multiple diseases There are ten anti-cancer/tumor-associated targets, namely MAPK10, GSK3B, APP, CASP7, CASP8, CSNK2A1, HSPA5, CASP3, MAPK1, and PRKCA. The MAPK signaling pathway has eight targets, namely MAPK10, HSPA8, CASP3, KDR, MAPK1, PRKCA, EGFR, and CDC25B.

## Network construction and analysis

The PKVG rhizome compound-target-pathway network contains 142 nodes. In Fig 4, 3 compounds are shown in blue, 97 targets are shown in purple, and 42 pathways are shown in red. The core targets were mined using the Centiscape 2.2 plug-in in Cytoscape 3.9.1 software. BC, CC, and DC of target proteins were calculated using topological analysis. The selection criteria were set as follows: BC > 260.59, CC > 0.0025, and DC > 4.52. As shown in Table 2, we

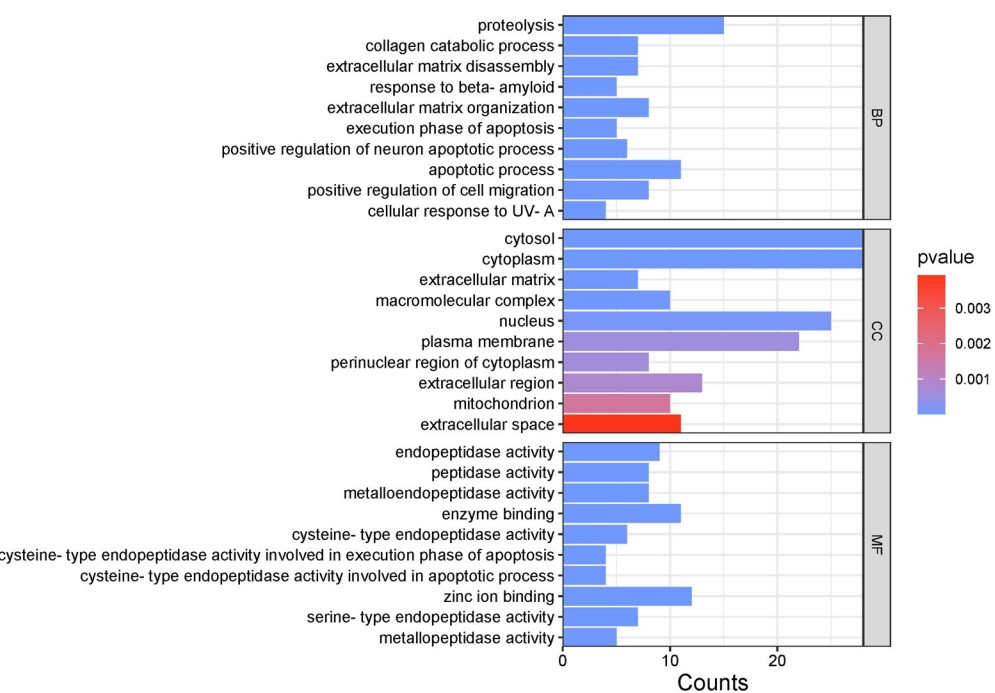

**Fig 2. Results of GO pathway enrichment analysis for anti-cancer/tumor activity of phenolic acids from PKVG rhizomes.**

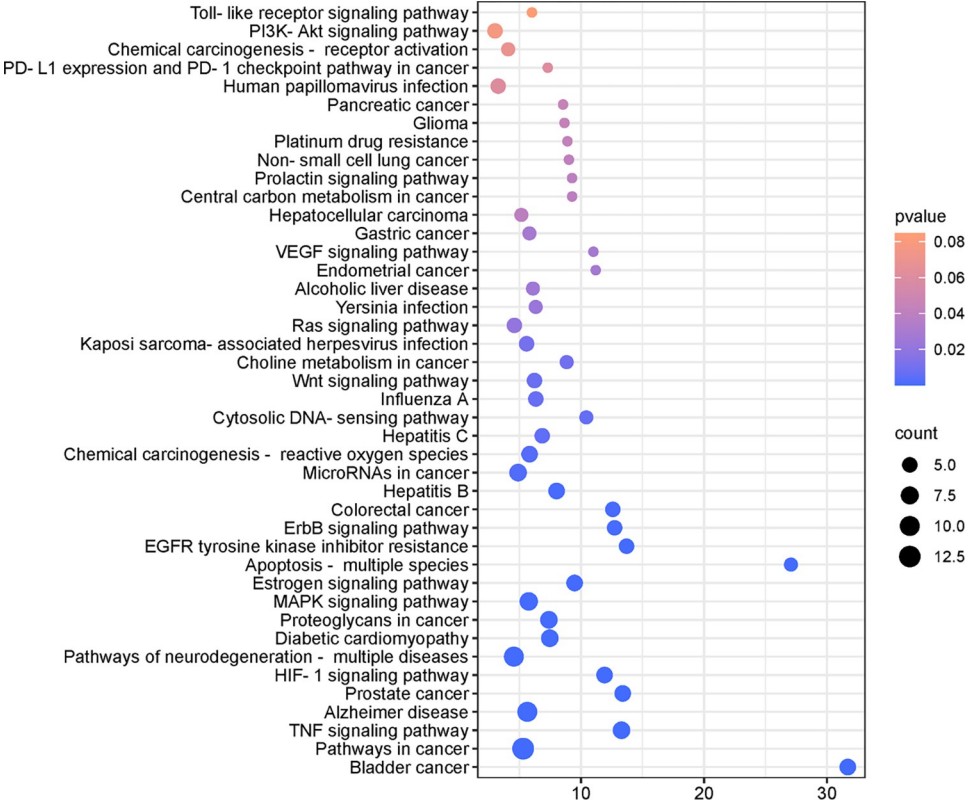

**Fig 3. Results of KEGG pathway enrichment analysis for anti-cancer/tumor activity of phenolic acids from PKVG rhizomes.** Bubble color and size correspond to the p-value and gene number enriched in the pathway.

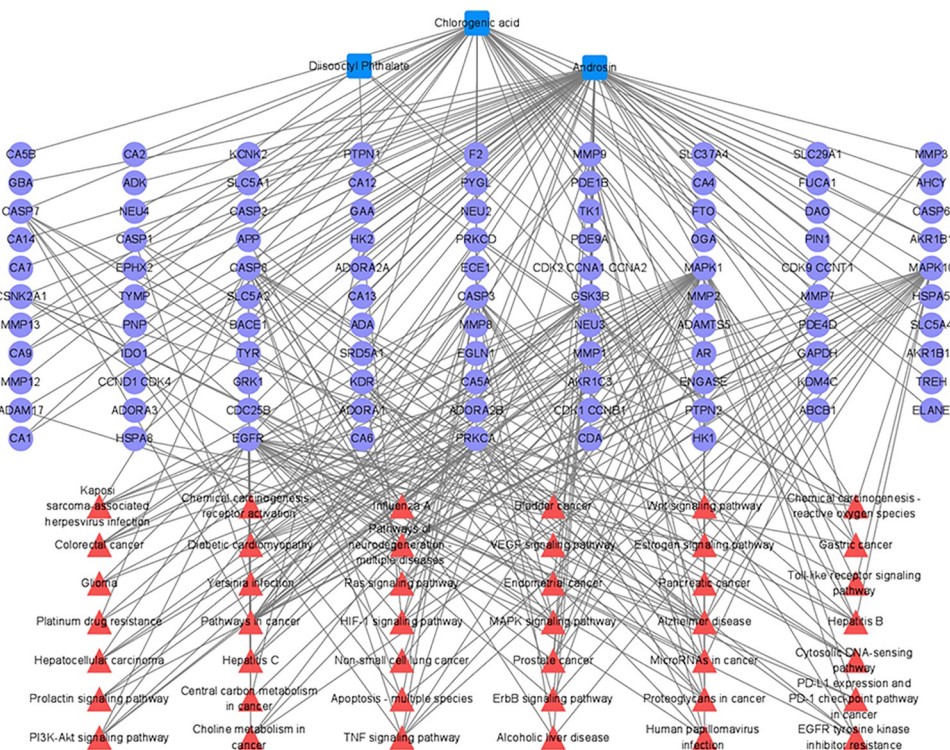

**Fig 4. Compound-target-pathway network for anti-cancer/tumor activity of PKVG rhizome phenolic acid analogs.** Blue squares represent active components in the PKVG rhizome, purple circles represent predicted targets, and red triangles represent associated pathways.

obtained 2 key compounds and 7 above-average values for the key targets. The four targets with the highest degree values among all targets were mitogen-activated protein kinase 1 (MAPK1; DC = 38, BC = 3016.51, CC = 0.0034), epidermal growth factor receptor (EGFR; DC = 27, BC = 1552.31, CC = 0.0031), and mitogen-activated protein kinase 10 (MAPK10; DC = 21, BC = 1177.56, CC = 0.0030) associated with androsin. While classical protein kinase

**Table 2. Network analysis results of key active ingredients, key targets and key pathways for anti-cancer/tumor activity of PKVG rhizome phenolic acid analogs.**

| No. | Name | Betweenness Centrality (BC) | Closeness Centrality (CC) | Degree Centrality (DC) |
|---|---|---|---|---|
| 1 | Androsin | 11308.37 | 0.0039 | 62 |
| 2 | Chlorogenic acid | 6314.03 | 0.0031 | 38 |
| 3 | MAPK1 | 3016.51 | 0.0034 | 38 |
| 4 | EGFR | 1552.31 | 0.0031 | 27 |
| 5 | PRKCA | 1129.64 | 0.0027 | 21 |
| 6 | MAPK10 | 1177.56 | 0.0030 | 21 |
| 7 | GSK3B | 1020.74 | 0.0030 | 20 |
| 8 | CASP3 | 789.40 | 0.0026 | 18 |
| 9 | CASP8 | 599.82 | 0.0026 | 15 |
| 10 | MMP9 | 383.00 | 0.0028 | 10 |
| 11 | Pathways in cancer | 769.10 | 0.0028 | 13 |
| 12 | Alzheimer disease | 342.20 | 0.0027 | 10 |
| 13 | Pathways of neurodegeneration—multiple diseases | 325.84 | 0.0027 | 10 |

**Table 3. Energy docking results of key targets and key phenolic acid compounds from PKVG rhizome.**

| Protein name | Gene name | Ligand name | Binding energy (kcal/mol) |
|---|---|---|---|
| MAP kinase ERK2 | MAPK1 | Androsin | -6.1 |
| Epidermal growth factor receptor erbB1 | EGFR | | -5.8 |
| c-Jun N-terminal kinase 3 | MAPK10 | | -7.8 |
| Protein kinase C alpha | PRKCA | Chlorogenic acid | -6.8 |

C alpha type (PRKCA; DC = 21, BC = 1129.64, CC = 0.0027) is associated with chlorogenic acid.

## Molecular docking results

The key compounds androsin and chlorogenic acid were molecularly docked to the four core targets with the highest degree of centrality: MAPK1, EGFR, MAPK10, and PRKCA. The binding energies are shown in Table 3. During molecular docking, the Androsin-MAPK1 docking parameter center (x, y, z) was (-4.433, 8.76, 47.404), the Androsin-EGFR docking parameter center (x, y, z) was (23.581, 9.73, 58.824), the Androsin-MAPK10 docking parameter center (x, y, z) was (9.249, 19.879, 23.573), and the Chlorogenic acid-PRKCA docking parameter center (x, y, z) was (5.254, 2.591, -0.453). All sizes (x, y, z) were (126, 126, 126). In order to evaluate the binding ability of key compounds and key targets, we used the empirical threshold (-5.0 kcal/mol) mentioned in the literature as an evaluation criterion, and if the docking binding energy was below the threshold, it indicated that the target had a strong binding ability to the compounds. The results showed that the binding ability of these key compounds to the four key targets was above the empirical threshold. We further analyzed the compound-target interactions (Fig 5). The results showed that Androsin had hydrogen bonds with all three targets (MAPK1, EGFR, and MAPK10), such as 3 hydrogen bonds in androsin-MAPK1, 4 hydrogen bonds in androsin-EGFR, androsin-MAPK10 has 6 hydrogen bonds, and chlorogenic acid-PRKCA has 7 hydrogen bonds. The presence of these types of bonds is the main reason why the binding energies of the complexes are all below the threshold. These results are also consistent with the results of the KEGG pathway enrichment analysis.

## Discussion

With the growing interest in the study of traditional Chinese herbs, many bioactive compounds have shown great pharmacological, biological, and medicinal value, including potential anticancer activities [17]. Cancer is currently a difficult challenge for mankind to overcome, and its occurrence is a complex process involving multiple steps in the transformation of normal cells into uncontrolled, proliferating cancer cells, a process commonly referred to as carcinogenesis or tumorigenesis [18]. In recent years, the active ingredients in many Chinese herbal medicines have been investigated using network pharmacology and molecular docking, and many targets for cancer/tumor treatment have been identified. For example, network pharmacology prediction and molecular docking of active ingredients in *Salvia miltiorrhiza* have been performed, and SRC, IL6, and INS were found to be associated with colorectal cancer [19]. Network pharmacological prediction and molecular docking of active ingredients in *Astragalus membranaceus* revealed that 1,7-dihydroxy-3,9-dimethoxy pterocarpene and isoflavanone may be the main active ingredients in the treatment of glioma [20]. Network pharmacological prediction and molecular docking of active ingredients from *Angelica dahurica* revealed that sen-byakangelicol, beta-sitosterol, and prangenin have high therapeutic osteosarcoma patients and improved osteosarcoma patients' five-year survival rates [21]. All of the

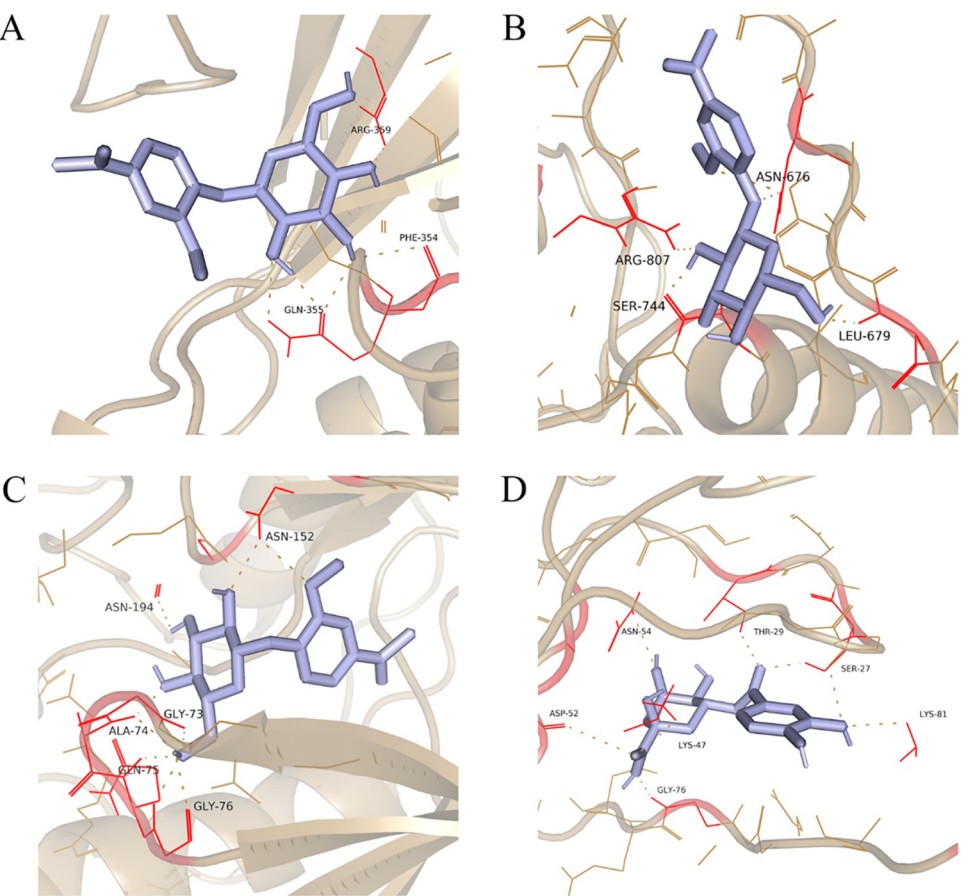

**Fig 5. Interaction graphics between the compounds and targets.** (A) Androsin-MAPK1, (B) Androsin-EGFR, (C) Androsin-MAPK10, (D) Chlorogenic acid-PRKCA.

above studies demonstrate the important role of network pharmacology prediction and molecular docking technology in the study of cancer therapy.

In this study, we identified 71 phenolic acids in PKVG rhizomes using broadly targeted metabolomics techniques. Three active ingredients (diisooctyl phthalate, androsin, and chlorogenic acid) were obtained by searching through the TCMSP database. These constituents with good medicinal properties are the material basis for the treatment of cancer/tumor with phenolic acids from PKVG rhizomes. There are 42 intersecting targets of cancer/tumor with PKVG phenolic acid-based active ingredients, and the top four DC were MAPK1, EGFR, MAPK10, and PRKCA by network analysis. Among them, the expression of MAPK1 inhibits the proliferation of rectal cancer cells [22]. CSIG-03. EGFR ligand in EGFR-amplified glioblastoma was able to activate the tumor suppressor of EGFR, which resulted in the conversion of EGFR oncogenes to oncogenes [23]. The tumor suppressor miR-335-5p was able to inhibit gastric cancer by targeting MAPK10 [24]. However, upregulation of PRKCA may be an unfavorable factor that leads to lung adenocarcinoma [25]. Therefore, it is important to explore the relationship between these target genes and the active components of PKVG phenolic acids to understand the mechanism of PKVG treatment of cancer/tumor.

GO functional enrichment and KEGG pathway analysis were performed on PKVG and cancer/tumor common target genes. It was found that GO enrichment analysis predicted that the active ingredients of luteolin might exert their cancer/tumor therapeutic effects through

biological processes such as proteolysis, the collagen catabolic process, and extracellular matrix disassembly of the active ingredients of luteolin. The above study suggests that multiple biological processes mediating and participating are the key to the anti-cancer/tumor effects of herbal PKVG. In KEGG enrichment analysis, inhibition of the PI3K-Akt signaling pathway is essential to inhibit cancer/tumor development. Aberrant activation of this signaling pathway is one of the most frequent events in human cancer and serves to disconnect the control of cell growth, survival and metabolism from exogenous growth stimuli [26, 27]. MAPK signaling pathway may be involved in the carcinogenesis and development of human meningiomas by binding to HER-2 [28], and inhibition of MAPK signaling pathway inhibition also suppresses glioma development [29]. Blocking the HIF-1 signaling pathway can prevent and treat cancer [30]. Moreover, the KEGG-enriched IL-17 signaling pathway [31] and TNF signaling pathway [32], are closely associated with cancer development.

We identified androsin and chlorogenic acid as the key herbal active ingredients in the rhizomes of PKVG through the TCMSP database. Previous studies have shown that androsin treatment not only reduces lipid levels in mouse hepatocytes and serum but also reduces HFrD-induced alanine aminotransferase (ALT), aspartate transaminase (AST), and cholesterol, resulting in the improvement of liver health in mice [33]. In addition, androsin has defensive effects against allergens and PAF-induced bronchial obstruction in guinea pigs [34]. However, so far, no study has demonstrated the inhibitory effect of androsin on cancer/tumor. However, in the present study, we found that androsin may have some therapeutic effects on cancer/tumor after medicinal target screening and molecular docking. Chlorogenic acid can regulate the glycolysis of tumor cells under hypoxic conditions [35], and its loaded chitosan nanoparticles also have tumor-preventive and antioxidant efficacy in experimental skin carcinogenesis [36]. The above result is similar to the predicted results of the present study.

## Conclusion

The phenolic acid chemical constituents in PKVG rhizomes were qualitatively and quantitatively analyzed using broadly targeted metabolomics techniques. From this, we identified 71 compounds; we then used a network pharmacology approach for target identification, pathway analysis, and network construction. Using this approach, the material basis and molecular mechanism of the anti-cancer/tumor effects of phenolic acids in PKVG rhizomes were explained for the first time to the best of our knowledge. Two key active components and eight key targets were obtained, and 42 major pathways were identified by KEGG pathway analysis. This finding reflects the multi-component, multi-target, and multi-pathway characteristics of traditional Chinese medicine.

## Author Contributions

**Funding acquisition:** Qiang Xiao.

**Investigation:** Lingjun Cui.

**Writing – original draft:** Xiaolin Wan.

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
