## [Decision Letter · Decision Letter 0]

17 Sep 2024

PONE-D-24-28940Metabolomics and network pharmacology-based identification of phenolic acids in Polygonatum kingianum var. grandifolium rhizomes as anti-Cancer/Tumor active ingredientsPLOS ONE

Dear Dr. Xiao,

Thank you for submitting your manuscript to PLOS ONE. After careful consideration, we feel that it has merit but does not fully meet PLOS ONE’s publication criteria as it currently stands. Therefore, we invite you to submit a revised version of the manuscript that addresses the points raised during the review process.

In this manuscript, authors have designed a study to explore the Metabolomics and network pharmacology-based identification of phenolic acids in Polygonatum kingianum var. grandifolium rhizomes as anti-Cancer/Tumor active ingredients. The findings of submitted manuscript suggest that that the phenolic acids in PKVG rhizomes may exert anti-cancer/tumor activity through a multi-component, multi-target, and multi-channel mechanism of action It is very well written and designed manuscipt for the human welfare however based on the learned reviewers’ recommendation, the submitted manuscript in its current form is not acceptable for publication in the esteemed “Plos One” journal and requires minor revision. It is suggested that authors must revise the manuscript as per reviewer’s comments and can resubmit for the publication in Plos One journal.   

We look forward to receiving your revised manuscript.

Kind regards,

Pankaj Singh, Ph.D.

Academic Editor

PLOS ONE

Journal Requirements:

5. Please amend your list of authors on the manuscript to ensure that each author is linked to an affiliation. Authors’ affiliations should reflect the institution where the work was done (if authors moved subsequently, you can also list the new affiliation stating “current affiliation:….” as necessary).

Reviewers' comments:

Reviewer's Responses to Questions

**Comments to the Author**

1. Is the manuscript technically sound, and do the data support the conclusions?

Reviewer #1: Yes

Reviewer #2: Yes

2. Has the statistical analysis been performed appropriately and rigorously? 

Reviewer #1: Yes

Reviewer #2: N/A

3. Have the authors made all data underlying the findings in their manuscript fully available?

Reviewer #1: Yes

Reviewer #2: Yes

4. Is the manuscript presented in an intelligible fashion and written in standard English?

Reviewer #1: Yes

Reviewer #2: Yes

5. Review Comments to the Author

Reviewer #1: Methods are not described in enough detail, and finally, Formatting inconsistencies throughout the manuscript, such as citation styles and section headings, could detract from its overall professionalism. Addressing these section-wise issues is crucial for enhancing the manuscript's quality and its chances of acceptance for publication.

In the other sections its been explained properly and the funding details if given in detail it could be better

Reviewer #2: The manuscript technically sound, and the data support the conclusions. The conclusions drawn appropriately based on the data presented. The manuscript presented in clear and comprehensible fashion and written in standard English.

6. PLOS authors have the option to publish the peer review history of their article (what does this mean?). If published, this will include your full peer review and any attached files.

Reviewer #1: No

Reviewer #2: No

---

## [Author Response · Author response to Decision Letter 0]

20 Sep 2024

Dear Editor,

Thank you very much for kindly providing us with the opportunity to revise our manuscript entitled " Metabolomics and network pharmacology-based identification of phenolic acids in Polygonatum kingianum var. grandifolium rhizomes as anti-Cancer/Tumor active ingredients ". The reviewers’ comments are all valuable and very helpful for revising and improving our paper. The revised sections are highlighted in the revision. The main corrections in the paper and the response to the reviewer’s comments are as follows:

Response to Reviewers:

Reviewer #1:

1. Methods are not described in enough detail, and finally, Formatting inconsistencies throughout the manuscript, such as citation styles and section headings, could detract from its overall professionalism. Addressing these section-wise issues is crucial for enhancing the manuscript's quality and its chances of acceptance for publication. In the other sections, its been explained properly and the funding details, if given in detail, could be better

Response: We sincerely thank Reviewer #1 for the meticulous comment. Firstly, we provided a more detailed description of the methodology used in the article. Please refer to lines 111-186 of the revised manuscript for details. Secondly, we thoroughly revised the manuscript's chapter titles and citation formats to ensure consistency. Finally, we have added detailed funding information in lines 365–371 of the article. Thank you for your valuable comments.

Reviewer #2: 

1. The manuscript is technically sound, and the data support the conclusions. The conclusions drawn appropriately based on the data presented. The manuscript presented in clear and comprehensible fashion and written in standard English.

Response: We sincerely thank Reviewer #2 for the meticulous comment. To ensure its accuracy, we have made every effort to revise the article's grammar content. Thank you for your valuable comments.

If there are any other issues with our manuscript, please feel free to contact us at any time. We would greatly appreciate it.

---

## [Decision Letter · Decision Letter 1]

17 Nov 2024

PONE-D-24-28940R1Metabolomics and network pharmacology-based identification of phenolic acids in Polygonatum kingianum var. grandifolium rhizomes as anti-Cancer/Tumor active ingredientsPLOS ONE

Dear Dr. Xiao,

Thank you for submitting your manuscript to PLOS ONE. After careful consideration, we feel that it has merit but does not fully meet PLOS ONE’s publication criteria as it currently stands. Therefore, we invite you to submit a revised version of the manuscript that addresses the points raised during the review process.

Authors have addressed each queries very well rose by reviewers and have critically modify the manuscript as per requirements but we have received two review reports from other reviewers which have some minor concern regarding the facts. By resolving the queries, raised by the reviewer, will help further improvement in the revised manuscript. Hence, It is suggested that authors must revise the manuscript as per reviewer’s comments and can resubmit for the publication in Plos One journal.       

We look forward to receiving your revised manuscript.

Kind regards,

Pankaj Singh, Ph.D.

Academic Editor

PLOS ONE

Journal Requirements:

Reviewers' comments:

Reviewer's Responses to Questions

**Comments to the Author**

1. If the authors have adequately addressed your comments raised in a previous round of review and you feel that this manuscript is now acceptable for publication, you may indicate that here to bypass the “Comments to the Author” section, enter your conflict of interest statement in the “Confidential to Editor” section, and submit your "Accept" recommendation.

Reviewer #3: (No Response)

Reviewer #4: (No Response)

2. Is the manuscript technically sound, and do the data support the conclusions?

Reviewer #3: No

Reviewer #4: Yes

3. Has the statistical analysis been performed appropriately and rigorously? 

Reviewer #3: N/A

Reviewer #4: Yes

4. Have the authors made all data underlying the findings in their manuscript fully available?

Reviewer #3: Yes

Reviewer #4: Yes

5. Is the manuscript presented in an intelligible fashion and written in standard English?

Reviewer #3: No

Reviewer #4: Yes

6. Review Comments to the Author

Reviewer #3: In this study, the aim is to investigate the phenolic acids in Polygonatum kingianum var. grandifolium rhizomes as anti-Cancer/Tumor active ingredients based on metabolomics and network pharmacology. All in all, the approach of this work is relatively conventional and lacks necessary cell or animal experiments for validation. There are still some suggestions for reference:

1. Authors are requested to double-check the grammar of the entire text.

2. How to identify metabolites needs to be clearly stated in the method.

3. What is the basis for screening criteria OB ≥ 5% and DL ≥ 0.14?

4. The topic of this work is too broad; it is recommended to choose one type of cancer or tumor for research.

5. Please add molecular docking coordinates and BOX information used in present work.

6. “C18” 18 Subscript required, please verify similar specialized expressions.

7. The discussion section also needs to go deeper in response to the results.

Reviewer #4: Please write the reviewer's recommendations and their response.

Legends are left with poor specifications.

7. PLOS authors have the option to publish the peer review history of their article (what does this mean?). If published, this will include your full peer review and any attached files.

Reviewer #3: No

Reviewer #4: No

---

## [Author Response · Author response to Decision Letter 1]

19 Nov 2024

Reviewer #3:

1. Authors are requested to double-check the grammar of the entire text. 

Response: We have done our best to correct the grammatical issues in the article.

2. How to identify metabolites needs to be clearly stated in the method. 

Response: We added how to identify metabolites in the Materials and Methods section. See lines 112-132 of the revised draft for details.

3. What is the basis for screening criteria OB ≥ 5% and DL ≥ 0.14?

Response: Although the commonly used criteria are OB ≥ 30% and DL ≥ 0.1, this is not absolute. In a study on astragalus-albizia medicine at the First Clinical Hospital of Shandong University of Traditional Chinese Medicine, researchers used OB ≥ 10% and DL ≥ 0.1 as the screening criteria and found that IL-1β is a potentially key mechanism for the treatment of chronic obstructive pulmonary disease (COPD). Therefore, we also used this criterion for the Chinese herb Polygonatum kingianum var. grandifolium. We have checked this section and cited the relevant literature in the manuscript. See line 142 and reference 16 of the revised manuscript for details.

4. The topic of this work is too broad; it is recommended to choose one type of cancer or tumor for research. 

Response: The aim of our study is to get the active components and targets of Polygonatum kingianum var. grandifolium phenolic acid analogs related to anti-cancer or tumors so as to lay the foundation for future research in our group. Therefore, our screening scope is broad. Other members of our group are conducting network pharmacological studies and animal studies on specific cancers or tumors based on existing research data. Therefore, we would like to publish this research article that analyzes a wide range of studies.

5. Please add molecular docking coordinates and BOX information used in present work. 

Response: We have added the relevant information. See lines 270-275 of the revised draft for details.

6. “C18” 18 Subscript required, please verify similar specialized expressions. 

Response: We have scrutinized and revised the contents of the manuscript. See Table 1 of the revised manuscript for details.

7. The discussion section also needs to go deeper in response to the results. 

Response: We provide a more in-depth discussion of the results section. See the discussion section of the revised draft for more details.

Reviewer #4:

1. Legends are left with poor specifications.

Response: We have revised the content of the legend. See the revised draft for details.

If you have any questions about our manuscript, please feel free to contact us and we will get back to you as soon as possible.

---

## [Editor Report · Decision Letter 2]

2 Dec 2024

Metabolomics and network pharmacology-based identification of phenolic acids in Polygonatum kingianum var. grandifolium rhizomes as anti-Cancer/Tumor active ingredients

PONE-D-24-28940R2

Dear Dr. Xiao,

We’re pleased to inform you that your manuscript has been judged scientifically suitable for publication and will be formally accepted for publication once it meets all outstanding technical requirements.

Kind regards,

Pankaj Singh, Ph.D.

Academic Editor

PLOS ONE
---

## [Editor Report · Acceptance letter]

6 Dec 2024

PONE-D-24-28940R2 

PLOS ONE

Dear Dr. Xiao, 

I'm pleased to inform you that your manuscript has been deemed suitable for publication in PLOS ONE. Congratulations! Your manuscript is now being handed over to our production team.

Kind regards, 

on behalf of

Dr. Pankaj Singh 

Academic Editor

PLOS ONE